# Thermally Controllable Decolorization of Reusable Radiochromic Complex of Polyvinyl Alcohol, Iodine and Silica Nanoparticles (PAISiN) Irradiated with γ-rays

**Hiroshi Yasuda** [1,*] and **Hirokazu Miyoshi** [2]

1 Department of Radiation Biophysics, Research Institute for Radiation Biology and Medicine, Hiroshima University, Hiroshima 734-8553, Japan

2 Advance Radiation Research, Education, and Management Center, Tokushima University, Tokushima 770-8501, Japan; miyoshi.hirokazu@tokushima-u.ac.jp

* Correspondence: hyasuda@hiroshima-u.ac.jp

**Featured Application: A reusable radiochromic complex composed of polyvinyl alcohol, iodine and silica nanoparticles, named "PAISiN", has preferable features for on-site real-time monitoring of hand exposure for medical and industrial workers.**

**Abstract:** Some medical and industry workers using ionizing radiation sources have potential risks of accidental high-dose exposure of their extremities, particularly their hands. While practical dosimeters suitable for on-site real-time monitoring of hand exposure are not yet available, they are desirable to be developed. Thus, the authors focused on the application of a reusable radiochromic complex composed of polyvinyl alcohol, iodide and silica nanoparticles, named "PAISiN", and examined their dose responses and thermal stabilities of radiochromic reactions. Three PAISiN samples each were irradiated with 5, 10 and 20 Gy of [137]Cs γ-rays, and time changes of the radiation-induced colors were observed at different temperatures: 20 °C (in a laboratory), 40 °C (in an oven) and 5.5 °C (in a refrigerator). It was confirmed that the PAISiN samples presented a red color that was easily detectable by the naked eyesight immediately after irradiation. The coloration was cleared within 24 h for 5 Gy irradiation at room temperature. The decolorization process was remarkably accelerated at 40 °C; it was erased in just 2 h. In contrast, storing in the refrigerator (5.5 °C) kept the color persistently for at least 4 days. These findings indicate that we could flexibly control the decolorization process of PAISiN in accordance with the objective of radiation monitoring.

**Keywords:** radiochromic; polyvinyl alcohol; iodine; silica nanoparticles; reusable; gel dosimeter; occupational exposure; hand monitoring

## 1. Introduction

Some medical and industry workers using radiation and radioisotopes have potential risks of accidentally receiving notably high doses to their extremities, particularly to their hands [1–5]. The dose received by a hand tends to be higher than the personal dose, which is generally monitored with a portable dosimeter attached to the chest or abdomen. Sometimes the hand dose exceeds the equivalent dose limit for skin (500 mSv per year) recommended by the International Commission on Radiological Protection (ICRP) [6], even though the personal dose is at an insignificant level [2,4]. Additionally, precise assessment of the non-uniform exposure of extremities is difficult since the dose distribution of a local body part could vary significantly depending on the type of radiation source, energy range of radiation and also the worker's posture, which can change with time in a complex manner. A novel technique to solve these issues is desirable to be developed.

For this aim, it is worth investigating radiochromic substances that are instantly colored following ionization induced by radiation. By putting a small piece of such

material to the fingertips, a worker using any radiation source could continuously monitor her/his hand exposure in real time and prevent excessive radiation exposure as necessary by stopping work soon after detecting a change of its color. A material employed for this objective at a variety of workplaces needs to satisfy the following requirements:

- Good usability: The monitor should be small, lightweight, robust and easily manageable, so that the work would be undisturbed;
- High detectability: Radiation-induced color change should be easily recognized soon after exposure of a few Gy (i.e., threshold dose level of major deterministic effects on human health);
- Stability: It should be stable in terms of dosimetric properties for a month (i.e., the general period for recording individual doses of workers);
- Safety: It should be non-hazardous for both humans and the biosphere;
- Sustainability: It should be reusable so that it would not produce waste.

With consideration of these requirements, the authors surveyed relevant studies on radiochromic materials that were developed and tested based on similar concepts [7–30]. As a consequence, the authors focused on a complex composed of polyvinyl alcohol (PVA), iodide (I) and silica nanoparticles (SiNP), named "PAISiN" here. This complex was developed and investigated with 30 kV X-rays by Miyoshi et al. [17]. They found that the addition of SiNP as a reductant worked as a radiation sensitizer for enhancing the radiochromic reaction. They also observed that the color of PAISiN changed from white to red immediately after irradiation with >0.5 Gy X-rays, and the coloration reverted to the initial state in 24 h by the reducing action of SiNP. The distinct color change was considered to be more easily detected by the naked eye than other radiochromic materials using ferric ions or dye. In addition, its reusability was thought to be desirable in views of both practicality and sustainability. It is known that similar radiochromic materials having the PVA-I matrix show notable natural coloration under room temperature due to auto-oxidation [27,29], which would be a critical shortcoming for routine monitoring of personal doses which are performed on a monthly basis in general. Furthermore, this complex has an economic advantage since it can be made of inexpensive reagents and inorganic materials used in an ordinary laboratory environment. However, the dosimetric properties of PAISiN for commonly used forms of radiation such as $\gamma$-rays and the thermal stabilities of its coloration/decolorization processes are yet to be examined.

## 2. Materials and Methods

### 2.1. Formation of the Complex

First, a PVA-I solution was formed by mixing 520 mg PVA (polymerization degree: 500, average saponification: 88 mol%) and 1788 mg of potassium iodide (KI), both of which were purchased from Wako Pure Chemicals Co., Ltd. (Osaka, Japan), in 12 mL of distilled water. In reference to a preceding study [31], nanoparticles of $SiO_2$ (SiNP) with a diameter of 12 nm were extracted from a commercial product of SiNP-dispersed aqueous solution (LUDOX HS-40, Sigma-Aldrich Co., Ltd., St. Louis, MI, USA) by washing out repeatedly with distilled water, and then 12 mL of 20 wt% SiNP solution was made. According to the finding that the radiation sensitivity of the radiochromic reaction of PAISiN was exponentially enhanced with increasing SiNP concentration [17], 1 mL of the PVA-I solution and 2 mL of the SiNP solution were put into a transparent quartz cell with dimensions of $12.5 \times 12.5 \times 58.0$ mm$^3$ coupled with a screw-type cap and mixed well by careful shaking to achieve the final composition of 0.66 mmol L$^{-1}$, 0.3 mol L$^{-1}$ KI and 13 wt% SiNP, which is slightly acidic due to the contribution of the residual acetyl group of polyvinyl acetate. It is considered that PVA is absorbed onto the SiNP surface due to the hydrogen bonding between hydroxyl groups and those in PVA in this complex [32]. A total of 10 cells of PAISiN were prepared. Some of its physico-chemical properties are shown in Table 1.



**Table 1.** Physico-chemical properties of PAISiN.

| Density | 1.075 g cm$^{-3}$ |
|---|---|
| Elemental composition [1] | K:1, Si:1, C:2, H:6, I:1, O:4 |
| Effective atomic number | 38.8 |

[1] Indicated with a relative atom number of each element. Note that Si was dispersed as silica nanoparticles in the solution.

### 2.2. Methods for Irradiation

Three cells of the PAISiN complex were irradiated with 5, 10 and 20 Gy (for water) of $^{137}$Cs source γ-rays at a dose rate of 0.75 Gy min$^{-1}$ using a commercial irradiator (Gammacell 40 Exactor Low Dose Rate Irradiator, Best Theratronics Ltd., Ottawa, ON, Canada) as shown in Figure 1. The same 9 samples were repeatedly used, and 3 cells each were irradiated 4 times with the same doses: 5, 10 or 20 Gy; thus, the total dose reached 20 Gy, 40 Gy and 80 Gy, respectively. The 1st experiment (i.e., γ-ray irradiation followed by storage at 20 °C) started on 4 October 2021, the 2nd (40 °C) on 11 October 2021, the 3rd (5.5 °C) on 18 October 2021 and the 4th (at 20 °C) on 8 November 2021; the whole period of this series of experiments was nearly 40 days. Every experiment started after the complete decolorization was confirmed by the naked eye.

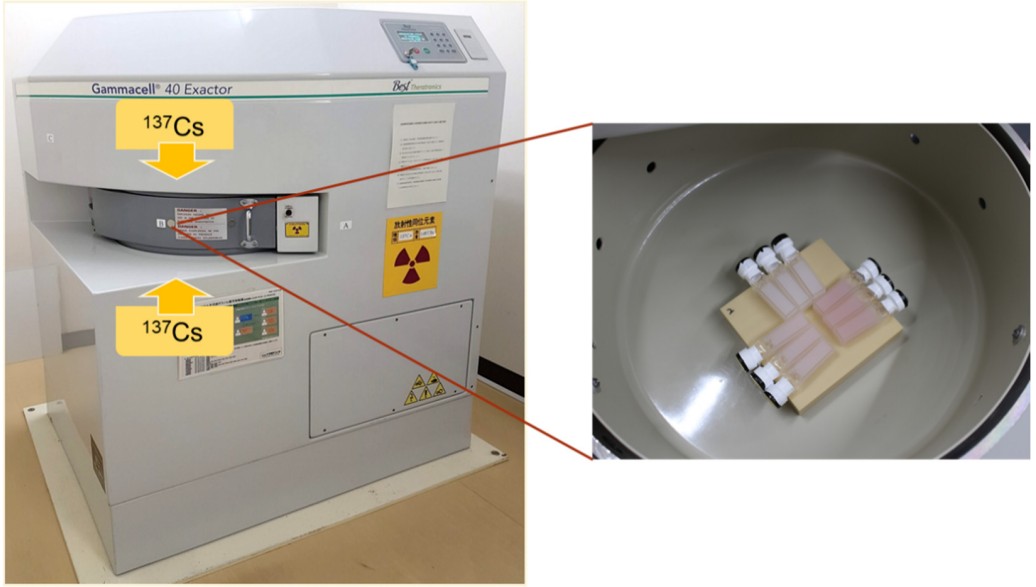

**Figure 1.** Experimental setup for irradiation using $^{137}$Cs source γ-rays; 3 samples each were irradiated with 5, 10 and 20 Gy (for water) with a dose rate of 0.75 Gy min$^{-1}$.

### 2.3. Post-Irradiation Analyses

The irradiated samples were stored at 3 different temperatures in the post-irradiation period: 20 °C in a laboratory room, 40 °C in an oven (MOV-112S, Sanyo Electric Biomedical Co., Ltd., Osaka, Japan) and 5.5 °C in a refrigerator. Afterward, irradiation followed by storage at 20 °C was performed again to check the reproducibility of the coloration/decolorization properties. Color images of the irradiated samples were taken using a flatbed scanner (GT-X900, Seiko Epson Corp., Nagano, Japan) with certain time intervals. The acquired images were analyzed using a freeware application, ImageJ, which is a Java-based image processing program developed by the National Institutes of Health and the Laboratory for Optical and Computational Instrumentation [33]. Measurements of color levels, color separations and calibrations were made by using the image processing functions of Image J. For examination of the dose responses of RGB components, the scanned color images were separated into three 8-bit images of blue, green and red components by using the built-in color split function of ImageJ.

## 3. Results and Discussions

### 3.1. Color Properties

The initial white color of PAISiN was changed to a red color immediately after γ-ray irradiation, as previously observed for 30 kV X-rays [17]. As examples, selected color images of PAISiN stored at room temperature (20 °C) for 0.25 h, 11.6 h and 70.6 h after irradiation are shown in Figure 2. It was confirmed that the pattern of its color change from clear white to nearly-pink red was easily detectable by the naked eye, which was preferable in a practical sense for on-site real-time radiation monitoring followed by flexible controls of occupational exposures.

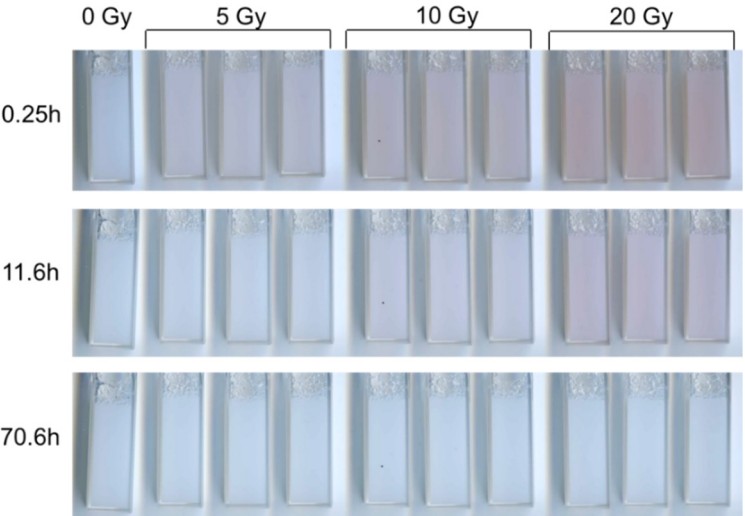

**Figure 2.** Scanned images of PAISiN stored at 20 °C taken at 0.25 h (**upper**), 11.6 h (**middle**) and 70.6 h (**lower**) after $^{137}$Cs γ-ray irradiation.

Figure 3 indicates the dose dependence of the 8-bit inverted intensities of the images shown in Figure 2 as a function of dose. It was seen that the radiation-induced coloration of PAISiN reverted to the initial white color in 3 days for this dose range (up to 20 Gy).

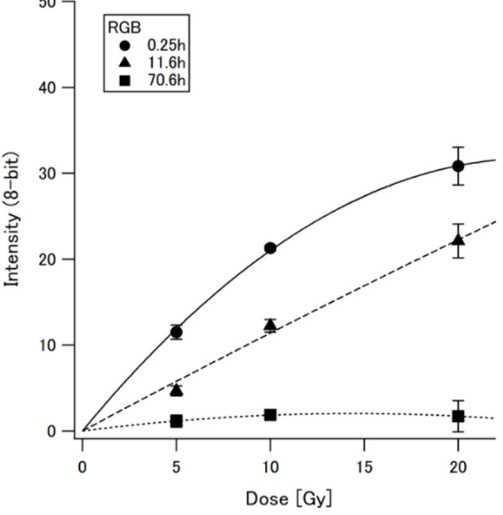

**Figure 3.** Dose responses of the 8-bit intensities of the inverted images of PAISiN at 0.25 h, 11.6 h and 70.6 h after γ-ray irradiation at room temperature (20 °C).

It should be noted that the radiation-induced coloration of PAISiN is more sensitive and more visible than that of the Fricke gel dosimeter. While this dosimeter shows radiation-induced coloration through the oxidation reaction from ferrous ($Fe^{2+}$) ions to ferric ($Fe^{3+}$)

ions, it needs tens of Gy to see a detectable color change [31]. Though gel-type Fricke dosimeters supplemented with xylenol orange, a $Fe^{3+}$ complexing agent, considerably increased its sensitivity [7–10,21–23,34], accurate color recognition by the naked eye is still challenging for low doses below 10 Gy, partially because of its indigenous orange color. PAISiN can present a visible red color which is detectable for an exposure of a few Gy γ-rays; PAISiN is expected to be more sensitive to diagnostic X-rays because of the larger contribution of photoelectric effects, as presumed from its high value of the effective atomic number (see Table 1). In addition, the decolorization process of PAISiN irradiated with 5 Gy can be complete in 1 day at room temperature, which is judged to be more advantageous for application to routine monitoring of occupational exposure compared to leuco-dye-based radiochromic materials, which need several days to revert to their initial color state [12,14,16,19].

### 3.2. Color Separation

According to the fact that a red color was induced by irradiation of PAISiN, it was presumed that the red component was less sensitive to radiation than other color components. Actually, it was reported that a higher-degree saponification of PVA lowered the absorbance of the red color (around 490 nm) [35,36], which implies that the red component of a PAISiN image is attributable to the formation of the residual acetyl group of polyvinyl acetate in PVA. With this consideration, the inverted images were split into three color components: blue, green and red; then, the dose responses of each component were examined using the same data of Figure 2. As seen in the results shown in Figure 4, it was confirmed that the intensities of the red components were notably lower than those of the blue and green components. While the blue component showed the highest intensity level, the variation at 20 Gy was found to be quite large. Based on these findings, it was decided to use the green component for further analyses in this study.

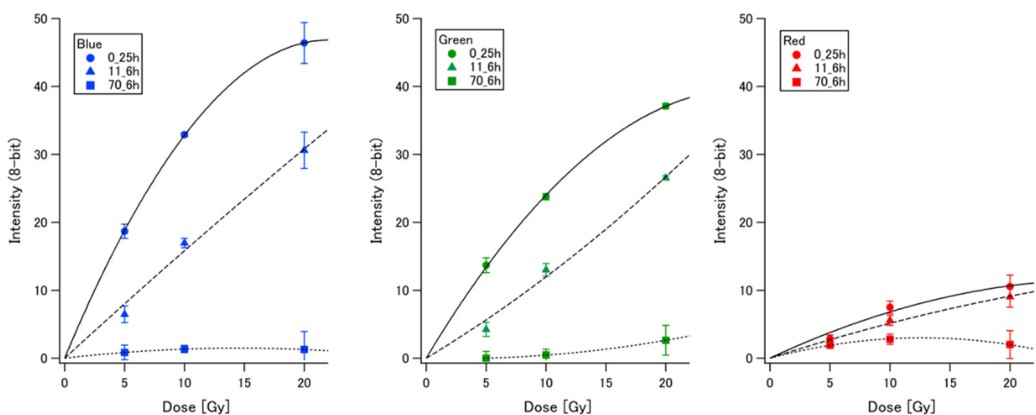

**Figure 4.** Dose responses of the inverted 8-bit intensities of blue (**left**), green (**middle**) and red (**right**) components of the PAISiN images acquired at room temperature (20 °C) at 0.25 h, 11.6 h and 70.6 h after γ-ray irradiation.

### 3.3. Thermal Effects on the Decolorization Process

Figure 5 shows the time changes of the green component intensities of PAISiN after irradiation with 5, 10 and 20 Gy γ-rays and being stored in a dark environment at 3 different temperatures: 20 °C (in a lab), 40 °C (in an oven) and 5.5 °C (in a refrigerator). The same set of PAISiN cells was irradiated repeatedly in this order after a complete decolorization was confirmed. As seen in the figures, the colors of PAISIN irradiated with 5 Gy returned to the initial white color within 24 h under room temperature (20 °C), while it took about 3 days to complete decolorization for 20 Gy irradiation. At 40 °C, the decolorization process was remarkably accelerated and complete in just 2 h, even for 20 Gy irradiation. In contrast, no decolorization was seen at 5.5 °C for at least 4 days; some enhancement of the color in the initial hours was observed, which implies that the PVA-I bonding was becoming

stronger with decreasing temperature. This implication is supported by the fact that the PVA-I matrix becomes a more stable equilibrium and maintains $I_2$ more firmly below 10 °C [37].

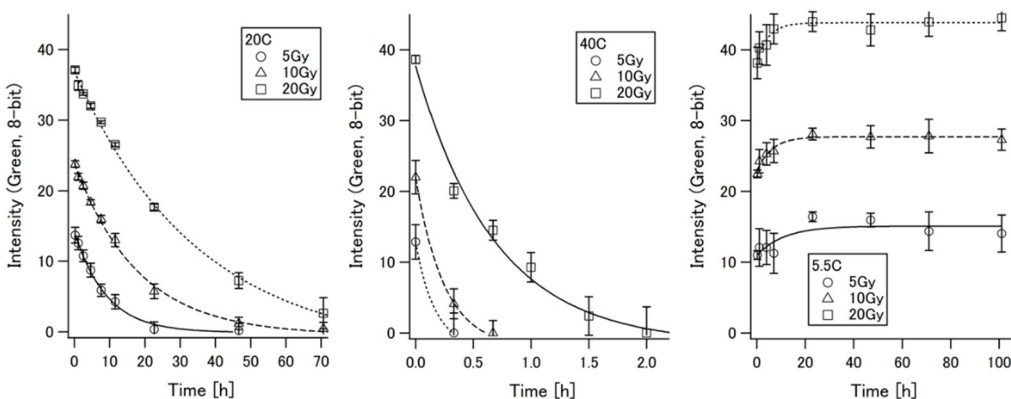

**Figure 5.** Time changes of the green component of PAISiN after irradiation of 5 Gy, 10 Gy or 20 Gy $\gamma$-rays under different temperatures: 20 °C (**left**), 40 °C (**middle**) and 5.5 °C (**right**).

Such a notable temperature dependence of the decolorization process of PAISiN could be explained by the acceleration or suppression of the chemical reactions triggered by ionizing radiation. The major processes of coloration and decolorization of PAISiN can be described as follows [16,26,38–41]:

$$\text{Radiolysis:} \quad H_2O \rightarrow OH^\bullet, HO_2^\bullet, H^\bullet, \text{etc.} \tag{1}$$

$$OH^\bullet + H^\bullet \rightarrow H_2O_2 \tag{2}$$

$$\text{Oxidation:} \quad H_2O_2 + I^- \rightarrow 0.5I_2 + OH^- + OH^\bullet \tag{3}$$

$$OH^\bullet + I^- \rightarrow HOI^- \tag{4}$$

$$HOI^- \rightarrow HO^- + I^\bullet \tag{5}$$

$$2I^\bullet + I^- \rightarrow I_3^- \tag{6}$$

$$\text{Complexation:} \quad I_3^- + PVA \rightarrow PVA\text{-}I_3^- \text{ (colored)} \tag{7}$$

$$\text{Dissociation:} \quad PVA\text{-}I_3^- \rightarrow 3I^- + PVA \text{ (decolorized)} \tag{8}$$

First, ionizing radiation induces the radiolysis of water ($H_2O$) and generates radicals such as $OH^\bullet$, $HO_2^\bullet$ and $H_2O_2$ (as indicated with Equations (1) and (2)). The rate constant (k) of the Equation (2) reaction is known to be k = $1.1 \times 10^{10}$ L mol$^{-1}$ s$^{-1}$ [38]. The same radicals are also made through the electrons and holes in SiNP after ionization [39,40]. Iodine ions ($I^-$) are oxidized by these radicals and changed to $I_3^-$ (Equations (3)–(6)). These $I_3^-$ molecules are bound to PVA [40] (Equation (7)), which is recognized as red color formation. The rate constant of the overall reaction between PVA and $OH^\bullet$ is $9.2 \times 10^{10}$ L mol$^{-1}$ s$^{-1}$ [41], while the reaction of $OH^\bullet$ and $I^\bullet$ is considered to be dominant because the molar concentration of $I^-$ (0.3 mol L$^{-1}$) was much higher than that of PVA (0.66 mmol L$^{-1}$). Under the condition of constantly receiving certain thermal energies (e.g., at room temperature), the PVA-$I_3^-$ gradually reverts to the iodine ions and PVA (Equation (8)) with time while losing the red color. This reverting, dissociating process is considered to occur on the surface of SiNP, which is negatively charged due to the deprotonated silanol group (-Si-O-) [42]. PVA is adsorbed onto the surface of SiNP through the hydrogen bonding between the hydroxyl groups of the SiNP surface and those in PVA [31]. $I_3^-$ bonds to the acetyl group in PVA adsorbed onto SiNP, and its dissociation is assumed to be accelerated at higher temperatures (e.g., in an oven) and suppressed at lower temperatures (e.g., in a refrigerator).

*3.4. Reproducibility of Radiation-Induced Coloration*

Though the radiation-induced red color of PAISiN was smoothly cleared and reverted to the initial white state at room or higher temperature, it was unclear whether its dosimetric properties were well recovered after repeated irradiation or long-term storage. Thus, after performing the experiments at 3 different temperatures (totally about 40 days after the PAISiN samples were synthesized), the same set of PAISiN cells were irradiated again with $^{137}$Cs source $\gamma$-rays in the same way (see the Materials and Methods section), and then time changes of the green component of the scanned PAISiN images were examined.

The results of the 2nd irradiations are shown in Figure 6. As recognized from a comparison of both graphs, the sensitivity of radiation-induced coloration notably reduced in the case of 20 Gy irradiation, while few changes were observed for 5 Gy irradiation. It was seen that the time needed for erasing the color became longer for all dose levels. These results indicate that some degradation of chemical properties of PAISiN in relation to coloration/decolorization occurred following repeated ionizations or thermal stimulations at certain temperatures. As the weight of any sample during about 1 month after the material synthesis was less than 0.5%, the increase of density caused by the loss of water was unlikely to have brought such changes in its dosimetric property. More studies to clarify the mechanism of these dose-dependent changes of sensitivity are desirable to be performed.

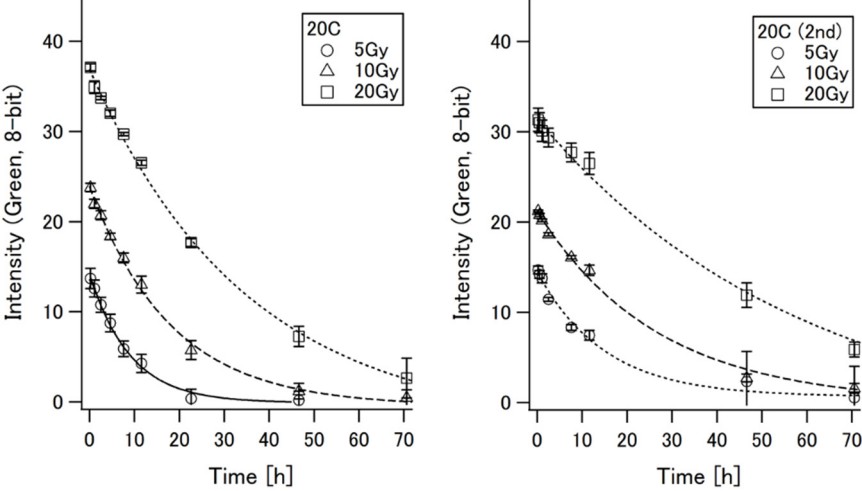

**Figure 6.** Comparison of the time changes of the green component of PAISiN at room temperature (20 °C) between repeated $\gamma$-ray irradiations: 1st (**left**) and 2nd (**right**); 2nd irradiation was made about 1 month later than the 1st one.

## 4. Conclusions

With the aim of application to on-site real-time monitoring for medical and industry workers whose extremities have potential risks of accidentally receiving high doses, the authors investigated the dose responses and thermal stabilities of a radiochromic material composed of PVA, iodine and silica nanoparticles, named "PAISiN". It was found that the radiation-induced coloration of PAISiN could be flexibly and simply controlled by the adjustment of temperature. We could quickly clear the color of PAISiN by heating around 40 °C for a few hours to reuse it; conversely, we could keep the radiation-induced color by storing it in a refrigerator for performing more comprehensive analyses later on. These findings add a practical advantage to PAISiN by highlighting its preferable features, such as high detectability, safety and cost-effectiveness.

For the application of PAISiN to routine monitoring in a variety of workplaces, further investigations to improve its accuracy, stability and reproducibility are desirable to be performed in reference to the preceding approaches using other materials [30–44]. Those studies should include experiments to examine the UV absorbance, higher dose-response,

longer-term stability, changes in dosimetric properties after more repetitive irradiation and preventive measures against degradation during storage, in parallel with comprehensive theoretical analyses on the thermally dynamic kinetics of coloration/decolorization and molecular interactions with varying compositions.

**Author Contributions:** Conceptualization, H.Y. and H.M.; material synthesis, H.M.; software setting and analysis, H.Y.; validation, H.Y. and H.M.; data curation, H.Y.; writing—original draft preparation, H.Y.; writing—review and editing, H.M.; funding acquisition, H.Y. All authors have read and agreed to the published version of the manuscript.

**Funding:** This research was funded by JSPS KAKENHI Grant Number 18KK0147 and the Program of the Network-type joint Usage/Research Center for Radiation Disaster Medical Science funded by the Ministry of Education, Culture, Sports, Science, and Technology (MEXT) of Japan and Hiroshima University.

**Institutional Review Board Statement:** Not applicable.

**Informed Consent Statement:** Not applicable.

**Data Availability Statement:** Not applicable.

**Conflicts of Interest:** The authors declare no conflict of interest.

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
