# Peer review of "Thermally Controllable Decolorization of Reusable Radiochromic Complex of Polyvinyl Alcohol, Iodine and Silica Nanoparticles (PAISiN) Irradiated with γ-rays"

_applsci, doi:10.3390/app12062959_

Round 1

Reviewer 1 Report

The paper presents an innovative method for the determination of the dose received at the extremities by visual observation. The paper is clear and methodical. Here are some comments.

Material and method section:

* could you clarify the role and function of Si nanoparticles in your study?

* could you clarify the conditions of the PVA-I and SiNP mixture? What do you mean by "blend" in line 83?

* concerning the triplicate repetition of the experiments, could you specify if these repetitions are carried out on the same tubes (for example, the tube irradiated at 5 Gy is irradiated 3 times in a row) or if you have prepared new samples for each repetition. In this case, how many times did you leave between two consecutive irradiations? If you prepared new samples, did you use the same initial mixture of PVA-I and SiNP?

* you varied several parameters: temperature, cumulative dose. Did you also perform experiments by varying the concentration of iodine, PVA and SiNP? If so, what were the results?

Result section:

* the results shown in Figures 3-6 are expressed as intensity (8-bit) versus dose or time. In order to fully understand your method, a section should be added in the "material and method" part on the processing of the data that allowed you to obtain these intensity values: how did you obtain these results? by which method? which software, if any?

* in the same vein, in section 3.2 you did a treatment by separating the blue-green-red colors: which method and software did you use?

* for the two previous comments, how do you calibrate to obtain an intensity value?

* in section 3.3, could you please clarify the mechanism of coloration and decoloration of the PVA-I complex? (1) indeed, in lines 182 and 192, you refer to 2 articles that do not correspond to the mechanisms you mention in equations 1 to 5 according to the references section on the one hand, and on the other hand reference 38 does not exist in the references list. (2) Could you please indicate the reaction constants of these reactions? For example, this would help to account for the reaction kinetics between reactions 2 and 3. (3) could you specify where the iodide ions are located in relation to the PVA structure? Indeed, you mention the formation of I2 by oxidation of I- followed by the formation of a PVA-I2 bond: I don't understand how the structure is arranged, and I can't make the connection with the methodology you indicate in section 2.1 - lines 77 to 86 on the formation of the PVA-I complex. There are missing steps or explanations to better understand the mechanism you propose (reactions 1 to 5). (4) since they are radicals, asterisks should be replaced by dots.

* in addition to your experiments, (1) UV absorbance measurements would be complementary to elucidate your mechanism. If we refer to the paper of Yokota and Kimura (1993), they show that two species I- and I3- can be formed (depending on the temperature) but the origin of their formation would be different. (2) to evaluate the limits of your device, did you perform higher irradiations (higher than 20 Gy)? if you did repetitions on the same tube, did you go beyond 3 repetitions, for example 5 or 10 repetitions? If so, what was the result?

Reviewer 2 Report

The article is written about the potential of dosimeter material due to colour changes post-irradiation. The report investigates the most sensitive radiochromic substance that can instantly change colour due to ionising radiation. However, there is no specific purpose that has been designed either for the industry, medical or for the environment. This is important to recognise the particular use of this potential radichromic substance in radiation dosimetry. As we know, the range of energy and radiation dose will differ for different purposes. 

It would be great to investigate extreme dose ranges to know the limit of radiation interaction. Since this interaction involves chemical response after irradiation, we need to know at what minimal dose the colour will change and at what the maxima dose it can be so that author may conclude the potential of this substance for medical and industrial workers as mentioned in line 226-228.

Results show that temperature contributes to the sensitivity of the radiochromic substance. So why did you choose 20 and 40 degrees Celcius?

How can you explain the chemical interaction in colour change post-irradiation and decolourisation at 40 degrees Celcius? What is the significance of each substance, such as iodine and silica nanoparticles?

Reviewer 3 Report

In this paper, the authors have developed PVOH, Iodine, and PAISiN nanoparticles composites for the monitoring of gamma radiation in the medical field. 

Following are some of the suggestions.

  1. Please provide some understanding about the dispersion of Iodine and PAISiN in PVOH polymer. You can study the molecular interaction between polymer and filler using FTIR and thermal analysis. The following papers will provide some understanding about how to characterize them. Also, comment on the mechanical properties of the composite film.
  • Bioresources, 2018,  13(2), 3195-3207
  • Journal of Polymer Science Part B: Polymer Physics 2015,53, 1519-1526.
  • International Journal of Adhesion and Adhesives 2017,78, 256.

2. Please comment on how the decoloration process impacts the thermal degradation of polymer composites. Up to how many cycles one can reuse the polymeric film. 

  •  
